# PAC-Bayes Analysis Beyond the Usual Bounds

**Omar Rivasplata**
University College London & DeepMind
o.rivasplata@cs.ucl.ac.uk

**Ilja Kuzborskij**
DeepMind
iljak@google.com

**Csaba Szepesvári**
DeepMind
szepi@google.com

**John Shawe-Taylor**
University College London
jst@cs.ucl.ac.uk

## Abstract

We focus on a stochastic learning model where the learner observes a finite set of training examples and the output of the learning process is a data-dependent distribution over a space of hypotheses. The learned data-dependent distribution is then used to make randomized predictions, and the high-level theme addressed here is guaranteeing the quality of predictions on examples that were not seen during training, i.e. generalization. In this setting the unknown quantity of interest is the expected risk of the data-dependent randomized predictor, for which upper bounds can be derived via a PAC-Bayes analysis, leading to PAC-Bayes bounds.

Specifically, we present a basic PAC-Bayes inequality for stochastic kernels, from which one may derive extensions of various known PAC-Bayes bounds as well as novel bounds. We clarify the role of the requirements of fixed 'data-free' priors, bounded losses, and i.i.d. data. We highlight that those requirements were used to upper-bound an exponential moment term, while the basic PAC-Bayes theorem remains valid without those restrictions. We present three bounds that illustrate the use of data-dependent priors, including one for the unbounded square loss.

## 1 Introduction

The context of this paper is the statistical learning model where the learner observes training data $S = (Z_1, Z_2, \ldots, Z_n)$ randomly drawn from a space of size-$n$ samples $\mathcal{S} = \mathcal{Z}^n$ (e.g. $\mathcal{Z} = \mathbb{R}^d \times \mathcal{Y}$ for a supervised learning problem where the input space is $\mathbb{R}^d$ and the label set is $\mathcal{Y}$) according to some unknown probability distribution[1] $P_n \in \mathcal{M}_1(\mathcal{S})$. Typically $Z_1, \ldots, Z_n$ are independent and share a common distribution $P_1 \in \mathcal{M}_1(\mathcal{Z})$. Upon observing the training data $S$, the learner outputs a *data-dependent* probability distribution $Q_S$ over a *hypothesis space* $\mathcal{H}$. Notice that this learning scenario involves randomness in the data and the hypothesis. In this stochastic learning model, the randomized predictions are carried out by randomly drawing a fresh hypothesis for each prediction. Therefore, we consider the performance of a probability distribution $Q$ over the hypothesis space: the expected *empirical loss* is $Q[\hat{L}_s] = \int_{\mathcal{H}} \hat{L}_s(h) Q(dh)$, i.e. the $Q$-average of the standard empirical loss $\hat{L}_s(h) = \hat{L}(h, s)$ defined as $\hat{L}(h, s) = \frac{1}{n} \sum_{i=1}^n \ell(h, z_i)$ for a fixed $h \in \mathcal{H}$ and $s = (z_1, \ldots, z_n)$, where $\ell : \mathcal{H} \times \mathcal{Z} \to [0, \infty)$ is a given loss function. Similarly, the expected *population loss* is $Q[L] = \int_{\mathcal{H}} L(h) Q(dh)$, i.e. the $Q$-average of the standard population loss $L(h) = \int_{\mathcal{Z}} \ell(h, z) P_1(dz)$ for a fixed $h \in \mathcal{H}$, where $P_1 \in \mathcal{M}_1(\mathcal{Z})$ is the distribution that generates one random example.

An important component of our development is formalizing "data-dependent distributions over $\mathcal{H}$" in a way that makes explicit their difference to fixed "data-free" distributions over $\mathcal{H}$.

**Data-dependent distributions as stochastic kernels.** A data-dependent distribution over the space $\mathcal{H}$ is formalized as a *stochastic kernel*[2] from $\mathcal{S}$ to $\mathcal{H}$, which is defined as a mapping[3] $Q : \mathcal{S} \times \Sigma_{\mathcal{H}} \to [0, 1]$ such that (i) for each $B \in \Sigma_{\mathcal{H}}$ the function $s \mapsto Q(s, B)$ is measurable; and (ii) for each $s \in \mathcal{S}$ the function $Q_s : B \mapsto Q(s, B)$ is a probability measure over $\mathcal{H}$. We write $\mathcal{K}(\mathcal{S}, \mathcal{H})$ to denote the set of all stochastic kernels from $\mathcal{S}$ to —distributions over— $\mathcal{H}$. We reserve the notation $\mathcal{M}_1(\mathcal{H})$ for the set of 'data-free' distributions over $\mathcal{H}$. Notice that $\mathcal{M}_1(\mathcal{H}) \subset \mathcal{K}(\mathcal{S}, \mathcal{H})$, since every 'data-free' distribution can be regarded as a constant kernel.

With the notation just introduced, $Q_S$ stands for the distribution over $\mathcal{H}$ corresponding to a randomly drawn data set $S$. The stochastic kernel $Q$ can be thought of as describing a randomizing learner. One well-known example is the *Gibbs* learner, where $Q_S$ is of the form $Q_S(dh) \propto e^{-\gamma \hat{L}(h,S)} \mu(dh)$ for some $\gamma > 0$, with $\mu$ a base measure over $\mathcal{H}$. Note that, besides randomized predictors, other prediction schemes may be devised from a learned distribution over hypotheses, as for instance ensemble predictors and majority vote predictors (see the related literature in Section 4 below).

A common question arising in learning theory aims to explain the generalization ability of a learner: how can a learner ensure a 'well-behaved' population loss? One way to answer this question is via upper bounds on the population loss, also called *generalization bounds*. Often the focus is on the *generalization gap*, which is the difference between the population loss and the empirical loss, and giving upper bounds on the gap. There are several types of generalization bounds we care about in learning theory, with variations in the way they depend on the training data $S$ and the data-generating distribution $P_n$. The classical bounds (such as VC-bounds) depend on neither. *Distribution-dependent* bounds are expressed in terms of quantities related to the data-generating distribution (e.g. population mean or variance) and possibly constants, but not the data in any way. These bounds can be helpful to study the behaviour of a learning method on different distributions— for example, some data-generating distributions might give faster convergence rates than others. Finally, *data-dependent* bounds are expressed in terms of empirical quantities that can be computed directly from data. These are useful for building and comparing predictors [Catoni, 2007], and also for "self-bounding" [Freund, 1998] or "self-certified" [Pérez-Ortiz et al., 2020] learning algorithms, which are learning algorithms that use all the available data to simultaneously provide a predictor and a risk certificate that is valid on unseen examples.

*PAC-Bayesian* inequalities allow to derive distribution- or data-dependent generalization bounds in the context of the stochastic prediction model discussed above. The usual PAC-Bayes analysis introduces a reference 'data-free' probability measure $Q^0 \in \mathcal{M}_1(\mathcal{H})$ on the hypothesis space $\mathcal{H}$. The learned data-dependent distribution $Q_S$ is commonly called a *posterior*, while $Q^0$ is called a *prior*. However, in contrast to Bayesian learning, the PAC-Bayes prior $Q^0$ acts as an analytical device and may or may not be used by the learning algorithm, and the PAC-Bayes posterior $Q_S$ is unrestricted and so it may be different from the posterior that would be obtained from $Q^0$ through Bayesian inference. In this sense, the PAC-Bayes approach affords an extra level of flexibility in the choice of distributions, even compared to generalized Bayesian approaches [Bissiri et al., 2016].

In the following, for any given $Q \in \mathcal{K}(\mathcal{S}, \mathcal{H})$ and $s \in \mathcal{S}$, we write $Q_s[\hat{L}_s] = \int \hat{L}_s(h) Q_s(dh)$ and $Q_s[L] = \int L(h) Q_s(dh)$ for the expected empirical loss and the expected population loss, respectively. The focus of PAC-Bayes analysis is deriving bounds on the gap between $Q_S[L]$ and $Q_S[\hat{L}_S]$. For instance, the classical result of McAllester [1999] says the following: For a fixed 'data-free' distribution $Q^0 \in \mathcal{M}_1(\mathcal{H})$, bounded loss function with range $[0, 1]$, stochastic kernel $Q \in \mathcal{K}(\mathcal{S}, \mathcal{H})$ and for any $\delta \in (0, 1)$, with probability at least $1 - \delta$ over size-$n$ random samples $S$:

$$Q_S[L] - Q_S[\hat{L}_S] \le \sqrt{\frac{1}{2n-1} \left( \mathrm{KL}(Q_S \| Q^0) + \log\left(\frac{n+2}{\delta}\right) \right)}. \tag{1}$$

$\mathrm{KL}(\cdot \| \cdot)$ stands for the Kullback-Leibler divergence[4] which is defined for two given probability distributions $Q, Q'$ over $\mathcal{H}$ as follows: $\mathrm{KL}(Q \| Q') = \int_{\mathcal{H}} \log\left(dQ/dQ'\right) dQ$, where $dQ/dQ'$ denotes the Radon-Nikodym derivative. Note that PAC-Bayes bounds (e.g. McAllester's bound described above) are usually presented under a statement that says that with probability at least $1 - \delta$, the

displayed inequality holds simultaneously for all probability distributions $Q$ over $\mathcal{H}$, i.e. with an arbitrary $Q$ replacing $Q_S$. Such commonly used formulation has the apparent advantage of being valid uniformly for all distributions over $\mathcal{H}$, while our formulation is valid for a fixed kernel. At the same time, the commonly used formulation has the disadvantage of hiding the data-dependence of the 'posterior' distributions used in practice, while our formulation in terms of a stochastic kernel shows explicitly the data-dependence: given the data $S$, the corresponding distribution over $\mathcal{H}$ is $Q_S$. Notice that one fixed stochastic kernel suffices in order to describe a whole parametric family of distributions (such as Gaussian or Laplace distributions, among others) with parameter values learned from data. Since our main interest is in results for data-dependent distributions (contrasted to results for fixed 'data-free' distributions), we argue in favour of the formulation based on stochastic kernels. These have appeared in the learning theory literature under the names of Markov kernels [Xu and Raginsky, 2017] or regular conditional probabilities [Catoni, 2004, 2007, Alquier, 2008].

A large body of subsequent work focused on refining the PAC-Bayes analysis by means of alternative proof techniques and different ways to measure the gap between $Q_S[L]$ and $Q_S[\hat{L}_S]$. For instance Langford and Seeger [2001] and Seeger [2002] gave an upper bound on the relative entropy of $Q_S[\hat{L}_S]$ and $Q_S[L]$, commonly called the PAC-Bayes-kl bound [Seldin et al., 2012], which holds with high probability over randomly drawn size-$n$ samples $S$:

$$\mathrm{kl}(Q_S[\hat{L}_S] \,\|\, Q_S[L]) \leq \frac{1}{n} \left( \mathrm{KL}(Q_S\|Q^0) + \log\left(\tfrac{n+1}{\delta}\right) \right) . \tag{2}$$

$\mathrm{kl}(\cdot\|\cdot)$, appearing on the left-hand side of this inequality, denotes the binary KL divergence, which is by definition the KL divergence between the Bernoulli distributions with the given parameters:

$$\mathrm{kl}(q\|q') = q \log(\frac{q}{q'}) + (1-q)\log(\frac{1-q}{1-q'}) \quad \text{ for } \ q,q' \in [0,1].$$

Inequality (2) is tighter than (1) due to Pinsker's inequality $2(p-q)^2 \leq \mathrm{kl}(p\|q)$. In fact, by a refined form of Pinsker's inequality, namely $(p-q)^2/(2q) \leq \mathrm{kl}(p\|q)$ which is valid for $p < q$ (and tighter than the former when $q < 0.25$), from Eq. (2) one obtains a *localised* inequality[5] (see Eq. (6) of McAllester [2003]), which holds with high probability[6] over randomly drawn size-$n$ samples $S$:

$$Q_S[L] - Q_S[\hat{L}_S] \lesssim \sqrt{\frac{Q_S[\hat{L}_S]}{n} \, \mathrm{KL}(Q_S\|Q^0)} + \frac{1}{n}\,\mathrm{KL}(Q_S\|Q^0) . \tag{3}$$

PAC-Bayes bounds like Eq. (1) and Eq. (3) tell us that the population loss is controlled by a trade-off between the empirical loss and the deviation of the posterior from the prior as captured by the KL divergence. Note that inequality (3) is tighter than (1) when $Q_S[\hat{L}_S] < Q_S[L] < 0.25$. Obviously, the upper bound in Eq. (3) is dominated by the lower-order (second) term whenever the empirical loss $Q_S[\hat{L}_S]$ is small enough, which makes this inequality very appealing for learning problems based on empirical risk minimization, where the empirical loss is driven to zero. At a high level, such kinds of data-dependent upper bounds on the generalization gap are much desirable, as their empirical terms are closely linked to—and hopefully capture more properties of—the data. In this direction, valuable contributions were made by Tolstikhin and Seldin [2013] who obtained an empirical PAC-Bayes bound similar in spirit to Eq. (3), but controlled by the sample variance of the loss. An alternative direction to get sharper empirical bounds was explored through *tunable* bounds [Catoni, 2007, van Erven, 2014, Thiemann et al., 2017], which involve a free parameter that offers a trade-off between the empirical error term and the $\mathrm{KL}(\text{Posterior}\|\text{Prior})$ term.

Despite their variety and attractive properties, the results discussed above (and the vast majority of the literature) share two crucial limitations: the prior $Q^0$ cannot depend on the training data $S$ and the loss function has to be bounded. It is conceivable that in many realistic situations the population loss is effectively controlled by the KL "complexity" term—indeed, in most modern learning scenarios (e.g. training deep neural networks) the empirical loss is driven to zero. At the same time, the choice of a fixed 'data-free' prior essentially becomes a wild guess on how the posterior will look like. Therefore, allowing prior distributions to be data-dependent introduces much needed flexibility, since this opens up the possibility to minimize upper bounds in both the posterior *and the prior*, which should lead to tighter empirical bounds on $Q_S[L]$ and tighter risk certificates.

These limitations have been removed in the PAC-Bayesian literature in special cases. For instance, Ambroladze et al. [2007] and Parrado-Hernández et al. [2012] used priors that were trained on a held-out portion of the available data, thus enabling empirical bounds with PAC-Bayes priors that are data-dependent, but independent from the training set. Priors that depend on the full training set have also been studied recently. Thiemann et al. [2017] proposed to construct a prior as a mixture of point masses at a finite number of data-dependent hypotheses trained on a $k$-fold split of the training set, effectively a data-dependent prior. Another approach was proposed by Dziugaite and Roy [2018b]: rather than splitting the training data, they require the data-dependent prior $Q_s^0$ (where $Q^0 \in \mathcal{K}(\mathcal{S}, \mathcal{H})$) to be stable with respect to 'small' changes in the composition of the $n$-tuple $s$. As we will see shortly, there is benefit in relaxing the restrictions of the usual PAC-Bayes literature.

## 2  Our Contributions

In this paper we discuss a basic PAC-Bayes inequality (Theorem 1 below) and a general template for PAC-Bayesian bounds (Theorem 2 below). The formulation of both these results is based on representing data-dependent distributions as stochastic kernels. To make a case for the usefulness of this approach, we show that our Theorem 2 encompasses many usual bounds which appear in the literature [McAllester, 1998, 1999, Seeger, 2002, Catoni, 2007, Thiemann et al., 2017], while at the same time it enables new PAC-Bayes inequalities. Importantly, our study takes a critical stand on the "usual assumptions" on which PAC-Bayes inequalities are based, namely, (a) data-free prior, (b) bounded loss, and (c) i.i.d. data observations. We aim to clarify the role of these assumptions and to illustrate how to obtain PAC-Bayes inequalities in cases where these assumptions are removed. As we will soon see, the analysis leading to our Theorem 2 shows that the PAC-Bayes priors can be data-dependent by default, and also that the underlying loss function can be unbounded by default. Furthermore, the proof of our Theorem 2 does not rely on the assumption of i.i.d. data observations, which may enable new results for statistically dependent data in future research.

For illustration, our general PAC-Bayes theorem[7] for stochastic kernels (Theorem 2 in Section 3), in specialized form, implies that for any convex function $F : \mathbb{R}^2 \to \mathbb{R}$, for any stochastic kernels $Q, Q^0 \in \mathcal{K}(\mathcal{S}, \mathcal{H})$ and $\delta \in (0, 1)$, with probability at least $1 - \delta$ over randomly drawn $S$ one has

$$F(Q_S[L], Q_S[\hat{L}_S]) \leq \mathrm{KL}(Q_S \| Q_S^0) + \log(\xi(Q^0)/\delta) , \qquad (4)$$

where $\xi(Q^0)$ is the exponential moment of $F(L(h), \hat{L}_s(h))$, which is defined as follows:

$$\xi(Q^0) = \int_{\mathcal{S}} \int_{\mathcal{H}} e^{F(L(h), \hat{L}_s(h))} Q_s^0(dh) P_n(ds) .$$

Observe that Eq. (4) is defined for an arbitrary convex function $F$. This way the usual bounds are encompassed: $F(x, y) = 2n(x - y)^2$ yields a McAllester [1999]-type bound, $F(x, y) = n \, \mathrm{kl}(y \| x)$ gives the bound of Seeger [2002], and $F(x, y) = n \log \left( \frac{1}{1 - x(1 - e^{-\lambda})} \right) - \lambda n y$ gives the bound of Catoni [2007]. Furthermore, $F(x, y) = n(x - y)^2 / (2x)$ leads to the so-called PAC-Bayes-$\lambda$ bound of Thiemann et al. [2017], or to the bound of Rivasplata et al. [2019] which holds under the usual requirements of fixed 'data-free' prior $Q^0$, losses within the $[0, 1]$ range, and i.i.d. data:

$$Q_S[L] \leq \left( \sqrt{Q_S[\hat{L}_S] + \frac{\mathrm{KL}(Q_S \| Q^0) + \log(\frac{2\sqrt{n}}{\delta})}{2n}} + \sqrt{\frac{\mathrm{KL}(Q_S \| Q^0) + \log(\frac{2\sqrt{n}}{\delta})}{2n}} \right)^2 . \qquad (5)$$

As consequence of the universality of Eq. (4), besides the usual bounds we may derive novel bounds, e.g. with data-dependent priors $Q_S^0$. Conceptually, our approach splits the usual PAC-Bayesian analysis into two components: (i) choose $F$ to use in Eq. (4), and (ii) obtain an upper bound on the exponential moment $\xi(Q^0)$. The cost of generality is that for each specific choice of the bound (technically, a choice of a function $F$ and $Q^0$) we need to study the exponential moment $\xi(Q^0)$ and, in particular, provide a reasonable, possibly data-dependent upper bound on it. We stress that the only technical step necessary for the introduction of a data-dependent prior is a bound on $\xi(Q^0)$, the rest is taken care of by Eq. (4). While previous works[8] analysed separately the exponential moment,

as we do here, to the best of our knowledge they considered data-free priors only. We think our work is the first to point out techniques to upper bound $\xi(Q^0)$ when $Q^0$ is a stochastic kernel, and to present PAC-Bayesian inequalities where the prior is data-dependent by default. Our work also clarifies where / how the data-free nature of the priors was used in previous works.

We emphasize that in this paper the main focus is on using data-dependent priors in the PAC-Bayes analysis. Again, we point out that the proof of the basic PAC-Bayes inequality (Theorem 1 below) does not require fixed 'data-free' priors, nor bounded loss functions nor i.i.d. data observations. The same can be said of Theorem 2, a consequence of Theorem 1(ii), which gives a general template for deriving PAC-Bayes bounds. Below we discuss three generalization bounds with data-dependent priors, two of which are for bounded losses, while the third is for the unbounded square loss.

## 2.1 A PAC-Bayes bound with a data-dependent Gibbs prior

Choosing as prior an *empirical Gibbs* distribution $Q_s^0(dh) \propto e^{-\gamma \hat{L}(h,s)}\mu(dh)$ for some fixed $\gamma > 0$ and base measure $\mu$ over $\mathcal{H}$, we derive a novel PAC-Bayes bound. Recall that $s$ is the size-$n$ sample. We use $F(x,y) = \sqrt{n}(x-y)$, and we prove that in this case the exponential moment $\xi(Q^0)$ satisfies

$$\log(\xi(Q^0)) \leq 2\left(1 + \frac{2\gamma}{\sqrt{n}}\right) + \log\left(1 + \sqrt{e}\right) .$$

The proof (Appendix B) is based on the algorithmic stability argument for Gibbs densities, inspired by the proof of Kuzborskij et al. [2019, Theorem 1]. Combining this with Eq. (4), for any kernel $Q \in \mathcal{K}(\mathcal{S}, \mathcal{H})$ and $\delta \in (0,1)$, with probability at least $1 - \delta$ over size-$n$ i.i.d. samples $S$ we have

$$Q_S[L] - Q_S[\hat{L}_S] \leq \frac{1}{\sqrt{n}}\left(\mathrm{KL}(Q_S\|Q_S^0) + 2\left(1 + \frac{2\gamma}{\sqrt{n}}\right) + \log\left(\frac{1 + \sqrt{e}}{\delta}\right)\right) . \qquad (6)$$

Notice that this prior allowed to remove '$\log(n)$' from the usual PAC-Bayes bounds (see our Eq. (1) and Eq. (2) above). This was one of the important contributions of Catoni [2007], who also used a data-dependent Gibbs distribution, see Catoni [2007, Theorem 1.2.4, Theorem 1.3.1, & corollaries]. Interestingly, the choice $Q = Q^0$ gives the smallest right-hand side in Eq. (6) (however, it does not necessarily minimize the bound on $Q_S[L]$) which leads to the following for the Gibbs learner: $Q_S[L] - Q_S[\hat{L}_S] \lesssim 1/\sqrt{n} + \gamma/n$ . Notice that this latter bound has an additive $1/\sqrt{n}$ compared to the bound in expectation of Raginsky et al. [2017].

## 2.2 PAC-Bayes bounds with d-stable data-dependent priors

Next we discuss an approach to convert any PAC-Bayes bound with a usual 'data-free' prior into a bound with a stable data-dependent prior, which is accomplished by generalizing a technique from Dziugaite and Roy [2018b]. Essentially, they show (see Appendix C) that for any fixed 'data-free' distribution $Q^* \in \mathcal{M}_1(\mathcal{H})$ and stochastic kernel $Q^0 \in \mathcal{K}(\mathcal{S}, \mathcal{H})$ satisfying the $\mathrm{DP}(\epsilon)$ property[9], one can turn the inequality $F(Q_S[L], Q_S[\hat{L}_S]) \leq \mathrm{KL}(Q_S\|Q^*) + \log(\xi(Q^*)/\delta)$ into

$$F(Q_S[L], Q_S[\hat{L}_S]) \leq \mathrm{KL}(Q_S\|Q_S^0) + \log(2\xi(Q^*)/\delta) + \frac{n\epsilon^2}{2} + \epsilon\sqrt{\frac{n}{2}\log(\frac{4}{\delta})} . \qquad (7)$$

In other words, if Eq. (4) holds with a data-free prior $Q^*$, then Eq. (7) holds with a data-dependent prior that is distributionally stable (i.e. satisfies $\mathrm{DP}(\epsilon)$). Note that different choices of $F$ would lead to different bounds on $\xi(Q^*)$ —essentially, upper bounds on the exponential moment typically considered in the PAC-Bayesian literature. For example, taking $F(x,y) = n\,\mathrm{kl}(y\|x)$ one can show that $\xi(Q^*) \leq 2\sqrt{n}$ [Maurer, 2004], and this leads to Theorem 4.2 of Dziugaite and Roy [2018b]: if $Q^0 \in \mathcal{K}(\mathcal{S}, \mathcal{H})$ satisfies the $\mathrm{DP}(\epsilon)$ property, then for any kernel $Q \in \mathcal{K}(\mathcal{S}, \mathcal{H})$ and $\delta \in (0,1)$, with probability at least $1 - \delta$ over size-$n$ i.i.d. samples $S$ we have

$$\mathrm{kl}(Q_S[\hat{L}_S]\|Q_S[L]) \leq \frac{1}{n}\left(\mathrm{KL}(Q_S\|Q_S^0) + \log(\frac{4\sqrt{n}}{\delta}) + \frac{n\epsilon^2}{2} + \epsilon\sqrt{\frac{n}{2}\log(\frac{4}{\delta})}\right) .$$

Eq. (7) is a general version of this result, whose derivation is based on the notion of *max-information* [Dwork et al., 2015a]. The details of the general conversion recipe are given in Appendix C.

## 2.3 A generalization bound for the square loss with a data-dependent prior

Our third and last contribution is a novel bound for the setting of learning linear predictors with the square loss. This will demonstrate the full power of our take on the PAC-Bayes analysis, as we will consider a regression problem with the unbounded squared loss and a data-dependent prior. In fact, our framework of data-dependent priors makes it possible to obtain the problem-dependent bound in Eq. (8) for square loss regression. We are not aware of an equivalent previous result.

In this setting, the input space is $\mathcal{X} = \mathbb{R}^d$ and the label space $\mathcal{Y} = \mathbb{R}$. A linear predictor is of the form $h_w : \mathbb{R}^d \to \mathbb{R}$ with $h_w(x) = w^\top x$ for $x \in \mathbb{R}^d$, where of course $w \in \mathbb{R}^d$. Hence $h_w$ may be identified with the weight vector $w$ and correspondingly the hypothesis space $\mathcal{H}$ may be identified with the weight space $\mathcal{W} = \mathbb{R}^d$. The size-$n$ random sample is $S = ((X_1, Y_1), \ldots, (X_n, Y_n)) \in (\mathbb{R}^d \times \mathbb{R})^n$. The population and empirical losses are defined with respect to the square loss function:

$$L(w) = \frac{1}{2} \mathbb{E}[(w^\top X_1 - Y_1)^2] \qquad \text{and} \qquad \hat{L}_S(w) = \frac{1}{2n} \sum_{i=1}^{n} (w^\top X_i - Y_i)^2 .$$

The population covariance matrix is $\boldsymbol{\Sigma} = \mathbb{E}[X_1 X_1^\top] \in \mathbb{R}^{d \times d}$ and its eigenvalues are $\lambda_1 \geq \cdots \geq \lambda_d$. The (regularized) sample covariance matrix is $\hat{\boldsymbol{\Sigma}}_\lambda = (X_1 X_1^\top + \cdots + X_n X_n^\top)/n + \lambda \boldsymbol{I}$ for $\lambda > 0$, with eigenvalues $\hat{\lambda}_1 \geq \cdots \geq \hat{\lambda}_d$. Note that $\hat{\lambda}_i$ are data-dependent.

Consider the prior $Q^0_{\gamma,\lambda}$ with density $q^0_{\gamma,\lambda}(w) \propto e^{-\frac{\gamma\lambda}{2}\|w\|^2}$ for some $\gamma, \lambda > 0$, that possibly depend on the data. In this setting, we prove (Appendix D) that for any posterior $Q \in \mathcal{K}(\mathcal{S}, \mathcal{W})$, for any $\gamma > 0$, and any $\lambda > \max_i\{\lambda_i - \hat{\lambda}_i\}$, with probability one over size-$n$ random samples $S$ we have

$$Q_S[L] - Q_S[\hat{L}_S] \leq \min_{w \in \mathbb{R}^d} L(w) + \frac{1}{\gamma} \text{KL}(Q_S \,\|\, Q^0_{\gamma,\lambda}) + \frac{1}{2\gamma} \sum_{i=1}^{d} \log\left(\frac{\lambda}{\lambda + \hat{\lambda}_i - \lambda_i}\right) . \qquad (8)$$

A straightforward observation is that this generalization bound holds *with probability one* over the distribution of size-$n$ random samples. This is a stronger result than usual high-probability bounds. Of course one may derive a high-probability bound from Eq. (8) by an application of Markov's inequality, but that would make the result weaker. The stronger result with probability one, for instance, allows to select the best out a countable collection of $\lambda$ values at no extra cost, while the high-probability bound would need to pay a union bound price for such selection.

Notice that we are not necessarily assuming bounded inputs or labels. Our bound depends on the data-generating distribution (possibly of unbounded support) via the spectra of the covariance matrices. While this is apparent by looking at the last term in Eq. (8), in fact the KL(Posterior‖Prior) term also depends on the covariances (see Proposition 12 in Appendix D). In particular, if the data inputs are independent sub-gaussian random vectors, then with high probability $|\hat{\lambda}_i - \lambda_i| \lesssim \sqrt{d/n}$ and the last term in Eq. (8) then behaves as $d \log\big(\lambda/(\lambda + \hat{\lambda}_i - \lambda_i)\big) \lesssim d/\sqrt{n-1}$. This of course can be extended to heavy-tailed distributions or, in general, to any input distributions such that spectrum of the covariance matrix concentrates well [Vershynin, 2011].

The explicit dependence on the spectrum of the sample covariance matrix opens interesting venues for distribution-dependent analysis. The above argument can be extended to heavy-tailed data distributions, where in some cases we can have concentration of the smallest eigenvalue of a sample covariance matrix even for unbounded instances, see Vershynin [2011, Section 5.4.2]. Moreover, our technique allows to combine PAC-Bayes analysis with specific applications by considering various data distributions. For instance, we can obtain bounds for structured data by analyzing eigenvalues of the corresponding (sparse or blocked) covariance matrices [Wainwright, 2019], thus revealing fined-grained dependence on the distribution compared to the usual PAC-Bayes bounds. Similarly, one can obtain generalization bounds for statistically dependent data by looking at the concentration of the covariance with dependent observations [de la Peña and Giné, 2012].

An important component of the proof of Eq. (8) is the following identity for the exponential moment of $f = \gamma(L(w) - \hat{L}_S(w))$ under the prior distribution: for $\lambda > \max_i\{\lambda_i - \hat{\lambda}_i\}$, with probability one over random samples $S$,

$$\log Q^0_{\gamma,\lambda}[e^f] = \gamma \min_{w \in \mathbb{R}^d} \left( L(w) - (\hat{L}_S(w) + \frac{\lambda}{2}\|w\|^2) \right) + \frac{1}{2} \sum_{i=1}^{d} \log\left(\frac{\lambda}{\lambda + \hat{\lambda}_i - \lambda_i}\right) . \qquad (9)$$

This identity computes explicitly the exponential moment of $f$ under the prior distribution. Also this explains why the upper bound in Eq. (8) contains the term $\min_{w \in \mathbb{R}^d} L(w)$. The latter should be understood as the label noise. This term will disappear in a noise-free problem, while given a distribution-dependent boundedness of the loss function, the term will concentrate well around zero (see Proposition 11 in Appendix D). We comment on the free parameter $\gamma$ in Appendix D.

Finally, note that Eq. (9) elucidates an equivalence between the concentration of eigenvalues of the sample covariance matrix and concentration of the empirical loss. Indeed, for simplicity assuming a noise-free setting (that is $\min_{w \in \mathbb{R}^d} L(w) = 0$), we observe that whenever $(\hat{\lambda}_i - \lambda_i) \to 0$ as $n \to \infty$ for i.i.d. instances, we have $\hat{L}_S(w) \to L(w)$. This provides an alternative way to control the concentration, compared to works based on restrictions on the loss as e.g. by Germain et al. [2016], Holland [2019]. We discuss another PAC-Bayes bound for unbounded losses in Appendix E.

## 3  Our PAC-Bayes theorem for stochastic kernels

The following results involve data- and hypothesis-dependent functions $f : \mathcal{S} \times \mathcal{H} \to \mathbb{R}$. Notice that the order $\mathcal{S} \times \mathcal{H}$ is immaterial—functions $\mathcal{H} \times \mathcal{S} \to \mathbb{R}$ are treated the same way. It will be convenient to define $f_s(h) = f(s, h)$. If $\rho \in \mathcal{M}_1(\mathcal{H})$ is a 'data-free' distribution, we will write $\rho[f_s]$ to denote the $\rho$-average of $f_s(\cdot)$ for fixed $s$, that is, $\rho[f_s] = \int_{\mathcal{H}} f_s(h)\rho(dh)$. When $\rho$ is data-dependent, that is, $\rho \in \mathcal{K}(\mathcal{S}, \mathcal{H})$ is a stochastic kernel, we will write $\rho_s$ for the distribution over $\mathcal{H}$ corresponding to a fixed $s$, so $\rho_s(B) = \rho(s, B)$ for $B \in \Sigma_{\mathcal{H}}$, and $\rho_s[f_s] = \int_{\mathcal{H}} f_s(h)\rho_s(dh)$.

The joint distribution over $\mathcal{S} \times \mathcal{H}$ defined by $P \in \mathcal{M}_1(\mathcal{S})$ and $Q \in \mathcal{K}(\mathcal{S}, \mathcal{H})$ is the measure denoted[10] by $P \otimes Q$ that acts on functions $\phi : \mathcal{S} \times \mathcal{H} \to \mathbb{R}$ as follows:

$$(P \otimes Q)[\phi] = \int_{\mathcal{S}} P(ds) \int_{\mathcal{H}} Q(s, dh)[\phi(s, h)] = \int_{\mathcal{S}} \int_{\mathcal{H}} \phi(s, h) Q_s(dh) P(ds) \,.$$

Drawing a random pair $(S, H) \sim P \otimes Q$ is equivalent to drawing $S \sim P$ and drawing $H \sim Q_S$. In this case, with $\mathbb{E}$ denoting the expectation under the joint distribution $P \otimes Q$, the previous display takes the form $\mathbb{E}[\phi(S, H)] = \mathbb{E}[\mathbb{E}[\phi(S, H)|S]]$. Our basic result is the following theorem.

**Theorem 1 (Basic PAC-Bayes inequality)** *Fix a probability measure $P \in \mathcal{M}_1(\mathcal{S})$, a stochastic kernel $Q^0 \in \mathcal{K}(\mathcal{S}, \mathcal{H})$, and a measurable function $f : \mathcal{S} \times \mathcal{H} \to \mathbb{R}$, and let*

$$\xi = \int_{\mathcal{S}} \int_{\mathcal{H}} e^{f(s,h)} Q_s^0(dh) P(ds) \,.$$

*(i) For any $Q \in \mathcal{K}(\mathcal{S}, \mathcal{H})$, for any $\delta \in (0, 1)$, with probability at least $1 - \delta$ over the random draw of a pair $(S, H) \sim P \otimes Q$ we have*

$$f(S, H) \leq \log \left( \frac{dQ_S}{dQ_S^0}(H) \right) + \log(\xi/\delta) \,.$$

*(ii) For any $Q \in \mathcal{K}(\mathcal{S}, \mathcal{H})$, for any $\delta \in (0, 1)$, with probability at least $1 - \delta$ over the random draw of $S \sim P$ we have*

$$Q_S[f_S] \leq \mathrm{KL}(Q_S \| Q_S^0) + \log(\xi/\delta) \,.$$

To the best of our knowledge, this theorem is new. Notice that $Q^0$ is by default a stochastic kernel from $\mathcal{S}$ to $\mathcal{H}$. Hence, given data $S$, the prior $Q_S^0$ is a data-dependent distribution over hypotheses. By contrast, the usual PAC-Bayes approaches assume that $Q^0$ is a 'data-free' distribution. Also note that the function $f$ is unrestricted, and the distribution $P \in \mathcal{M}_1(\mathcal{S})$ is unrestricted, except for integrability conditions to ensure that $\xi$ is finite. A key step of the proof involves a well-known change of measure that can be traced back to Csiszár [1975] and Donsker and Varadhan [1975].

**Proof** Recall that when $Y$ is a positive random variable, by Markov inequality, for any $\delta \in (0, 1)$, with probability at least $1 - \delta$ we have:

$$\log Y \leq \log \mathbb{E}[Y] + \log(1/\delta) \,. \tag{$\star$}$$

Let $Q^0 \in \mathcal{K}(\mathcal{S}, \mathcal{H})$, and let $\mathbb{E}^0$ denote expectation under the joint distribution $P \otimes Q^0$. Thus if $S \sim P$ and $H \sim Q_S^0$ we then have $\xi = \mathbb{E}^0[\mathbb{E}^0[e^{f(S,H)}|S]]$.

Let $Q \in \mathcal{K}(\mathcal{S}, \mathcal{H})$ and denote by $\mathbb{E}$ the expectation under the joint distribution $P \otimes Q$. Then by a change of measure we may re-write $\xi = \mathbb{E}^0[e^{f(S,H)}]$ as $\xi = \mathbb{E}[e^{\tilde{f}(S,H)}] = \mathbb{E}[e^D]$ with

$$ D = \tilde{f}(S,H) = f(S,H) - \log\left(\frac{dQ_S}{dQ_S^0}(H)\right). $$

(i) Applying inequality $(\star)$ to $Y = e^D$, with probability at least $1 - \delta$ over the random draw of the pair $(S, H) \sim P \otimes Q$ we get $D \le \log \mathbb{E}[e^D] + \log(1/\delta)$.

(ii) Recall $f_S(H) = f(S, H)$. Notice that $\mathbb{E}[D|S] = Q_S[f_S] - \mathrm{KL}(Q_S \| Q_S^0)$. By Jensen inequality, $\mathbb{E}[D|S] \le \log \mathbb{E}[e^D|S]$. While from $(\star)$ applied to $Y = \mathbb{E}[e^D|S]$, with probability at least $1 - \delta$ over the random draw of $S \sim P$ we have $\log \mathbb{E}[e^D|S] \le \log \mathbb{E}[e^D] + \log(1/\delta)$. ∎

Suppose the function $f$ is of the form $f = F \circ A$ with $A : \mathcal{S} \times \mathcal{H} \to \mathbb{R}^k$ and $F : \mathbb{R}^k \to \mathbb{R}$ convex. In this case, by Jensen inequality we have $F(Q_s[A_s]) \le Q_s[F(A_s)]$ and Theorem 1(ii) gives:

**Theorem 2 (PAC-Bayes for stochastic kernels)** *For any $P \in \mathcal{M}_1(\mathcal{S})$, for any $Q^0 \in \mathcal{K}(\mathcal{S}, \mathcal{H})$, for any positive integer $k$, for any measurable function $A : \mathcal{S} \times \mathcal{H} \to \mathbb{R}^k$ and convex function $F : \mathbb{R}^k \to \mathbb{R}$, let $f = F \circ A$ and let $\xi = (P \otimes Q^0)[e^f]$ as in Theorem 1. Then for any $Q \in \mathcal{K}(\mathcal{S}, \mathcal{H})$ and any $\delta \in (0, 1)$, with probability at least $1 - \delta$ over the random draw of $S \sim P$ we have*

$$ F(Q_S[A_S]) \le \mathrm{KL}(Q_S \| Q_S^0) + \log(\xi/\delta). \tag{10} $$

This theorem is a general template for deriving PAC-Bayes bounds, not just with 'data-free' priors, but also more generally with data-dependent priors. Previous works (see Section 4 below) that presented similar generic templates for deriving PAC-Bayes bounds only considered data-free priors. We emphasize that a 'data-free' distribution is equivalent to a constant stochastic kernel: $Q_s^0 = Q_{s'}^0$ for all $s, s' \in \mathcal{S}$. Hence $\mathcal{M}_1(\mathcal{H}) \subset \mathcal{K}(\mathcal{S}, \mathcal{H})$, which implies that our Theorem 2 encompasses the usual PAC-Bayes inequalities with data-free priors in the literature.

Interestingly, our Theorem 2 is valid with any normed space instead of $\mathbb{R}^k$. This theorem extends the typically used case where $k = 2$ and $A = (L(h), \hat{L}(h, s))$, in which case the function of interest is $f(s, h) = F(L(h), \hat{L}(h, s))$, where $F : \mathbb{R}^2 \to \mathbb{R}$ is a convex function, but there are no restrictions on the loss function $\ell$ that is used in defining $L(h)$ and $\hat{L}(h, s)$. Hence Theorem 2 is valid for *any* loss function: convex or non-convex, bounded or unbounded. Notice also that our Theorem 2 holds for any $P \in \mathcal{M}_1(\mathcal{S})$, i.e. without restrictions on the data-generating process. In particular, our Theorem 2 holds without the i.i.d. data assumption, hence this theorem could potentially enable new generalization bounds for statistically dependent data. In Section 4 below we comment on some literature related to unbounded losses and non-i.i.d. data.

An important role is played by $\xi$, the exponential moment (moment generating function at 1) of the function $f$ under the joint distribution $P \otimes Q^0$. As discussed above in Section 2, there are essentially two main steps involved in obtaining a PAC-Bayesian inequality: (i) choose $F$ to use in Theorem 2, and (ii) upper-bound the exponential moment $\xi$. We emphasize that the "usual assumptions" on which PAC-Bayes bounds are based, namely, (a) data-free prior, (b) bounded loss, and (c) i.i.d. data, played a role only in the technique used for controlling $\xi$. This is because with a data-free $Q^0$ we may swap the order of integration:

$$ \xi = \int_{\mathcal{S}} \int_{\mathcal{H}} e^{f(s,h)} Q^0(dh) P(ds) = \int_{\mathcal{H}} \int_{\mathcal{S}} e^{f(s,h)} P(ds) Q^0(dh) =: \xi_{\mathrm{swap}}. $$

Then bounding $\xi$ proceeds by calculating or bounding $\xi_{\mathrm{swap}}$ for which there are readily available techniques for bounded loss functions and i.i.d. data (see e.g. Maurer [2004], Germain et al. [2009], van Erven [2014]). The bounds with data-dependent priors that we presented in Section 2 required different kinds of techniques to control the exponential moment, the details are in the appendices. To the best of our knowledge, ours is the first work to extend the PAC-Bayes analysis to stochastic kernels. This framework appears to be a promising theoretical tool to obtain new results. The three types of data-dependant priors discussed in Section 2 show the versatility of the approach. Deriving more cases of PAC-Bayes inequalities without the usual assumptions is left for future research.

# 4 Additional discussion and related literature

The literature on the PAC-Bayes learning approach is vast. We briefly mention the usual references McAllester [1999], Langford and Seeger [2001], Seeger [2002], and Catoni [2007]; but see also Maurer [2004], and Keshet et al. [2011]. Note that McAllester [1999] continued McAllester [1998] whose work was inspired by Shawe-Taylor and Williamson [1997]'s work on a PAC analysis of a Bayesian-style estimator. We acknowledge the tutorials of Langford [2005] and McAllester [2013], the mini-tutorial of van Erven [2014], and the primer of Guedj [2019]. Our Theorem 2 is akin to general forms of the PAC-Bayes theorem given before by Audibert [2004], Germain et al. [2009], and Bégin et al. [2014, 2016]. Our Theorem 1(i) is akin to the "pointwise" bound of Blanchard and Fleuret [2007], in that the bound holds over the random draw of data and hypothesis pairs.

There are many application areas that have used the PAC-Bayes approach, but there are essentially two ways that a PAC-Bayes bound is typically applied: either use the bound to give a risk certificate for a randomized predictor learned by some method, or turn the bound itself into a learning method by searching a randomized predictor that minimizes the bound. The latter is mentioned already by McAllester [1999], credit for this approach in various contexts is due also to Germain et al. [2009], Seldin and Tishby [2010], Keshet et al. [2011], Noy and Crammer [2014], Keshet et al. [2017], possibility among others. Recently, the use of the latter approach has also found success in training neural networks, see Dziugaite and Roy [2017, 2018b]. In fact, the recent resurgence of interest in the PAC-Bayes approach has been to a large extent motivated by the interest in generalization guarantees for neural networks. Langford and Caruana [2001] used McAllester [1999]'s classical PAC-Bayesian bound to evaluate the error of a (stochastic) neural network classifier. Dziugaite and Roy [2017] obtained numerically non-vacuous generalization bounds by optimizing the same bound. Subsequent studies (e.g. Rivasplata et al. [2019], Pérez-Ortiz et al. [2020]) continued this approach, sometimes with links to the generalization of stochastic optimization methods (e.g. London [2017], Neyshabur et al. [2018], Dziugaite and Roy [2018a]) or algorithmic stability.

A line of work related to connecting PAC-Bayes priors to data was explored by Lever et al. [2013], Pentina and Lampert [2014] and more recently by Rivasplata et al. [2018], who assumed that priors are *distribution-dependent*. In that setting the priors are still 'data-free' but in a less agnostic fashion (compared to an arbitrary fixed prior), which allows to demonstrate improvements for "nice" data-generating distributions. Data-dependent priors were investigated recently by Awasthi et al. [2020], who relied on tools from the empirical process theory and controlled the capacity of a data-dependent hypothesis class (see also Foster et al. [2019]). The PAC-Bayes literature does contain a line of work that investigates relaxing the restriction of bounded loss functions. A straightforward way to extend PAC-Bayes inequalities to unbounded loss functions is to make assumptions on the tail behaviour of the loss [Alquier et al., 2016, Germain et al., 2016] or its moments [Alquier and Guedj, 2018, Holland, 2019], leading to interesting bounds in special cases. Recent work has also looked into the analysis for heavy-tailed losses. For example, Alquier and Guedj [2018] proposed a polynomial moment-dependent bound with $f$-divergence replacing the KL divergence, while Holland [2019] devised an exponential bound assuming that the second moment of the loss is bounded uniformly across hypotheses. An alternative approach was explored by Kuzborskij and Szepesvári [2019], who proposed a stability-based approach by controlling the Efron-Stein variance proxy of the loss. Squared loss regression was studied by Shalaeva et al. [2020] who improved results of Germain et al. [2016] and also relaxed the data-generation assumption to non-iid data. It is worth mentioning the important work related to extending the PAC-Bayes framework to statistically dependent data, see e.g. Alquier and Wintenberger [2012] who applied Rio [2000]'s version of Hoeffding's inequality, derived PAC-Bayes bounds for non-i.i.d. data, and used them in model selection for time series.

As we mentioned in the introduction, besides randomized predictions, other prediction schemes may be derived from a learned distribution over hypotheses. Aggregation by exponential weighting was considered by Dalalyan and Tsybakov [2007, 2008], ensembles of decision trees were considered by Lorenzen et al. [2019], weighted majority vote by Masegosa et al. [2020], Germain et al. [2015]. This list is far from being complete. Finally, it is worth mentioning that the PAC-Bayesian analysis extends beyond bounds on the gap between population and empirical losses: A large body of literature has also looked into upper and lower bounds on the *excess risk*, namely, $Q_S[L] - \inf_{h \in \mathcal{H}} L(h)$, we refer e.g. to Catoni [2007], Alquier et al. [2016], Grünwald and Mehta [2019], Kuzborskij et al. [2019], Mhammedi et al. [2019]. The approach of analyzing the gap (for randomized predictors), which we follow in this paper, is generally complementary to such excess risk analyses.

## Broader Impact

We think this work will have a positive impact on the theoretical machine learning community. However, since this work presents a high-level theoretical framework, its direct impact on society will be linked to the particular user-specific applications where this framework may be instantiated.

## Acknowledgments and Disclosure of Funding

We warmly thank the anonymous reviewers for their valuable feedback, which helped us to improve the paper greatly. For comments on various early parts of this work we warmly thank Tor Lattimore, Yevgeny Seldin, Tim van Erven, Benjamin Guedj, and Pascal Germain. We warmly acknowledge the Foundations team at Deepmind, and the AI Centre at University College London, for providing friendly and stimulating work environments. Omar Rivasplata and Ilja Kuzborskij warmly thank Vitaly Feldman for interesting discussions and a fun table tennis game while visiting DeepMind.

Omar Rivasplata gratefully acknowledges DeepMind sponsorship for carrying out research studies on the theoretical foundations of machine learning and AI at University College London. This work was done while Omar was a research scientist intern at DeepMind.

Csaba Szepesvári gratefully acknowledges funding from the Canada CIFAR AI Chairs Program, the Alberta Machine Intelligence Institute (Amii), and the Natural Sciences and Engineering Research Council (NSERC) of Canada.

John Shawe-Taylor gratefully acknowledges support and funding from the U.S. Army Research Laboratory and the U. S. Army Research Office, and by the U.K. Ministry of Defence and the U.K. Engineering and Physical Sciences Research Council (EPSRC) under grant number EP/R013616/1.

## Footnotes

[1] We write $\mathcal{M}_1(\mathcal{X})$ to denote the family of probability measures over a set $\mathcal{X}$, see Appendix A.

[2]This is also called a *transition kernel* or *probability kernel*, a well-known concept in the literature on stochastic processes, see e.g. Kallenberg [2017], Meyn and Tweedie [2009], Ethier and Kurtz [1986].

[3]The space of size-$n$ samples $\mathcal{S}$ is equipped with a sigma algebra that we denote $\Sigma_{\mathcal{S}}$, and the hypothesis space $\mathcal{H}$ is equipped with a sigma algebra that we denote $\Sigma_{\mathcal{H}}$. For precise definitions see Appendix A.

[4]Also known as relative entropy, see e.g. Cover and Thomas [2006].

[5] For $x, b, c$ nonnegative, $x \leq c + b\sqrt{x}$ implies $x \leq c + b\sqrt{c} + b^2$.

[6] The notation $\lesssim$ hides universal constants and logarithmic factors.

[7]Generic PAC-Bayes theorems, similar in spirit to ours, have been presented before, e.g. by Audibert [2004], Germain et al. [2009], Bégin et al. [2014, 2016], but only with fixed 'data-free' priors.

[8]Audibert and Bousquet [2007], Alquier et al. [2016], among others, for the case of fixed 'data-free' priors.

[9]$\mathrm{DP}(\epsilon)$ stands for "differential privacy with $\epsilon$." See Appendix C for details on this property.

[10] The notation $P \otimes Q$ (see e.g. Kallenberg [2017]), used here for the joint distribution over $\mathcal{S} \times \mathcal{H}$ defined by $P \in \mathcal{M}_1(\mathcal{S})$ and $Q \in \mathcal{K}(\mathcal{S}, \mathcal{H})$, corresponds to what in Bayesian learning is commonly written $Q_{H|S} P_S$.

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
