[Reviews · NeurIPS 2020]

Review 1

Summary and Contributions: The paper presents a KL-based PAC-Bayes generalization bound for any convex function between the expected and empirical Gibbs risks. While the initial results are for data-dependent priors, the authors also provide a general way to change a PAC-Bayes bound from a data-free prior to a data-dependent prior, by assuming a differential privacy property on the distributions. The paper also presents an example of unbounded loss, i.e., squared loss in a regression problem.

Strengths: The theoretical results seem sound.

Weaknesses: I am unclear about novelty and relevance. (Please see my additional feedback below.)

Correctness: The main claims seem correct.

Clarity: The paper is mostly clear. Few comments to improve clarity: Line 167 "DP(\epsilon) property" should better refer to "\epsilon-differential privacy" in the main text for clarity. I could not believe that Equation 8 holds with probability one, until I noticed that \lambda_i and \hat{lambda_i} depend on S. I suggest using the notation \lambda^S_i and \hat{lambda^S_i} to make the dependence clear. I know that this is somewhat clarified later in Lines 203-207. Line 18 "(e.g. \mathcal{Z} = R^d \times \mathcal{Y})" does not explain what \mathcal{Y} is in the example.

Relation to Prior Work: Relation to prior work is partially addressed. (Please see my additional feedback below.)

Reproducibility: Yes

Additional Feedback: The authors discussed several prior works for unbounded losses in Lines 320-328. While previous results rely on concentration of the losses (e.g., bounded moments), the current result also relies on concentration of the covariance eigenvalues. Note that the loss 1/2 E[(w^T X - y)^2] = 1/2 w^T E[X X^T] w - E[X y]^T w + 1/2 E[y^2], and thus, the loss includes the covariance matrix E[X X^T]. Is there any relationship between the previous assumptions regarding concentration of the losses (e.g., bounded moments) and the concentration of the covariance eigenvalues? I am unclear about novelty and relevance since the authors provide only one example (regression). I know that the authors made an effort to explain throughout the paper and in Section 4. I will appreciate if the authors can elaborate more on novelty and relevance. For instance: What new results can we obtain now that were not possible before. What better rates can we obtain for problems where we already had results. === AFTER REBUTTAL I am a bit disappointed by the authors response: "We will add discussions about the new kinds of results one can obtain now that were not possible before, including rates where relevant." It would have been more beneficial to use the rebuttal to provide such a case. As the authors agree, there is a relationship between the concentration of the loss and the concentration of the covariance eigenvalues. This decreases the novelty of the example provided by the authors. Given the above, I lower my evaluation from 6 to 5.


Review 2

Summary and Contributions: The paper extends the PAC-Bayesian analysis of data-dependent priors.

Strengths: This work study very important aspects of the PAC-Bayes theory. The introduced general framework to study data-dependant prior appears to be a promising theoretical tool to obtain new results, The proposed specialization to three types of data-dependant prior shows the versatility of the approach.

Weaknesses: Some results should be discussed in more details. (1) One asset of the "classical" PAC-Bayes bounds is the fact that they are valid uniformly for all posterior Q (as for the bounds of Eq. 1, 2 and 3). It does not seem to be the case for Theorem 1(i). Am I right? And what about the other bounds of the paper? This point deserve to be properly mentioned for all results. (2) Theorems 1 and 2 proposes general theorems that are possibly data-dependant, but borrows similarities from previous works. Theorem 1(i) looks akin the PAC-Bayes "pointwise" bound of Blanchard and Fleuret ("Occam’s Hammer", COLT 2007), Theorems 1(ii) and 2 resemble to many PAC-Bayesian general statements of the literature cited elsewhere in the manuscript. The function «F» domain might be in R^k instead of R^2, but it does not seem to be used or discussed in the paper.

Correctness: The theoretical statements are sounds and the mathematical results seem to have been obtained rigorously. However, I haven't check all the details in supplementary material.

Clarity: The clarity can be improved. The first pages contain many important definitions in footnotes, which I find unpleasant to read. Space could be saved by shortening, or moving to appendix, the final "Additional discussion and related literature" section, that is a very general and mostly uninformative list of general PAC-Bayes references. This should also allow to comment more on the results (see above " Weaknesses" section).

Relation to Prior Work: Previous work are cited but not always in an oprtimal manner (see comments above).

Reproducibility: Yes

Additional Feedback: Minor comments: Line 84: I don't think "by a stronger form of Pinsker's inequality" is appropriate to say, as (p-q)^2/(2q) < 2(p-q)^2 for q > 0.25. Thus, it only provides tighter risk bounds in some appropriate scenarios. Line 502: Z_i -> Z'_1 Line 510: DP is used before being defined (next page). ================== Following the rebuttal and the discussion, I raised my score from 6 to 7 because, as mentioned by R3, "the distinction between distribution-dependent bounds and data-dependent bounds was often poorly understood in the literature", and the paper could help the community to go further in this direction. I strongly encourage the authors to carefully revise their paper to include all the changes they commit to do in their rebuttal. Note, however, that they must be careful regarding the claim "each theorem holds for an arbitrary kernel, so in that sense it holds 'for all posteriors' as per the usual formulation in the literature." I doubt that the bound is valid *uniformly* for all kernel, i.e., I doubt that the following holds: P( \forall kernel : risk < bound) < 1-\delta. In contrast, the usual PAC-Bayes bounds are valid *uniformly* "for all posterior": P( \forall posterior: risk < bound) < 1-\delta.


Review 3

Summary and Contributions: The paper starts with an introduction to PAC-Bayes bounds: 1) what they are useful for: to upper-bound the generalization error of randomized or aggregated predictors 2) the various type of bounds (data dependent, distribution dependent...) Then, the authors proposes a new key lemma (Theorem 1) in PAC-Bayes analysis. Under the "usual assumptions", that is, a) bounded loss b) data-free prior and c) i.i.d obserbations, this key lemma allows to recover known PAC-Bayes bounds by McAllester (1998), Seeger (2002), Catoni (2007)... However, the important point is that the lemma still holds without these assumptions: there is a key quantity in the upper bound, \xi(Q^0). In any setting where one is able to upper-bound \xi(Q^0), one obtains a new PAC-Bayes inequality. The authors apply Theorem 1 to data-dependent priors (that is, b) above is violated), and then, to least-square regression (where a) above is violated). Note that in the first case, they obtain known essentially known results: one by Catoni (2007), the other by Dziugaite and Roy (2018) -- with an improvement in the constants, though.

Strengths: PAC-Bayes bounds are known to be quite "difficult to read", e.g. the monograph by Catoni (2007) is very technical. Despite the recent regain of interest in PAC-Bayes bounds, a good and easy-to-follow introduction is still needed. The introduction (pages 1 to 3) is the clearest I've never read on the topic. All the quantities are defined rigorously, and explained intuitively. Moreover, the distiction between distribution-dependent bounds and data-dependent bounds was often poorly understood in the literature: the explanations by the authors is great. Both types of bounds are useful, distribution-dependent bounds showing that "some data-generating distributions might give faster convergence rates than others" while data-dependent bounds are to be seen as "a risk certificate that is valid on unseen examples". For this beautiful introduction, I want to first congratulate the authors. The main contribution is Theorem 1, that is new to my knowledge. It provides a unified approach to many PAC-Bayes bounds, and clarifies the role of the "usual assumptions" mentioned above. It also provides a way to avoid them and to derive new bounds.

Weaknesses: One can argue that, even in the quadratic loss and data dependent prior case, the results were essentially already known. So, the paper mostly gives a unified approach to prove existing results. Indeed, I would not say that the paper is groundbreaking. Even though, this criticism should be mitigated: 1) the constants in the bound by Dziugaite and Roy (2018) are improved. Even when the improvement in the constants is small, I think this is always important in data-dependent bounds to obtain the sharpest possible constants. 2) the approach proposed by the authors can be applied to other examples, and lead to newer bounds.

Correctness: The main arguments in the proofs seem to be correct.

Clarity: As already written above (in the "strength" section), the clarity level of the paper is excellent.

Relation to Prior Work: Overall, the references to prior works is very good. The authors made a great effort to mention all classical PAC-Bayes works: McAllester (1998), Seeger (2002), Catoni (2007)... and to mention the most recent works (2018/2019), explaining well the contributions of each. I would like to raise 4 important points, though. I hope that the authors can take them into account: 1) The introduction (line 106-115) mention many works with data-dependent priors, but do no mention Catoni (2007). However, the fact that Catoni used data-dependent priors is menioned later, in Section 2.1. It might be worth mentioning it in the introduction, too. One more thing about data-dependent priors: the application of PAC-Bayes bounds with data-free priors, like (1), (2) and (3), leads to bounds in sqrt(log(n)/n) in classification. Using the Gibbs data-dependent prior, one obtains (6), where the bound is in sqrt(1/n). Thus, one gets rid of the log(n) term, which is to my opinion one of the most important contributions in Catoni (2007). Given the pedagogical qualities of the paper, it would be worth mentioning this explicitely. 2) I mentioned above 3 "classical assumptions": a) bounded loss b) data-free prior and c) i.i.d obserbations. The authors claim that their work allows to extend over past results by removing a) or b), but they never mention c). If one checks the proofs in [Alquier and Wintenberger, Model selection for weakly dependent time series forecasting, Bernoulli, 2012], it appears that the proofs are based on PAC-Bayes bounds for non i.i.d data. Namely, the authors apply Rio's version of Hoeffding's inequality for time series, derive PAC-Bayes bounds, and use them to prove model selection results for time series. The interesting point is that: I) Theorem 1 does not require assumption a) nor b), but it does not either require assumption c)!! and II) one can plug Rio's inequality to upper bound \xi(Q^0) when the data is not independent. To mention this would considerably extend the scope of the paper. 3) The authors write "To the best of our knowledge, ours is the first work to extend PAC-Bayes analysis to stochastic kernels". This statement is wrong. In Catoni (2007), the notion of kernel is defined page 5/6 and the following PAC-Bayes bound, Theorem 1.2.1, is stated for a stochastic kernel. (Note that there, it's not called a "kernel" but a "regular conditional probability measure", but the definition is exactly the same). The same framework was used for example in [Alquier, PAC-Bayesian Bounds for Randomized Empirical Risk Minimizers, Mathematical Methods of Statistics, 2008] Definition 1.5 page 282. 4) In Section 2.3, the authors study least-square regression and provide a very general result. However, line 205/206, it seems that one has to require that the noise is sub-Gaussian to obtain explicit rates. The authors mention Vershynin (2011) to remove this assumption, is it possible to have more details? I think another possibility is to use the work of Holland (2019), already cited by the authors. The idea of Holland is to use in the proofs an alternative loss function, that allows to deal with heavy-tailed noise. My guess is that the authors could use Holland's idea in their Theorem 1, by adapting the function f(S,H) so that it matches the loss function of Holland. If this is the case, this would be another interesting application of Theorem 1. (Of course, I understand that space is limited, so I'm not asking for complete statements of new results, but at least a short discussion on these points would be nice).

Reproducibility: Yes

Additional Feedback: None. ************** I thank the authors for their detailed feedback. My positive opinion on the paper remains unchanged, so I don't change my scores...


Review 4

Summary and Contributions: The paper handles PAC-Bayes bounds where the prior is data-dependent, or when the loss function is unbounded. The authors use the proposed method to provide several examples where this bound can be useful.

Strengths: The paper develops new bounds, some of them eliminate previous assumptions, thus allowing more holistic bounds. Specifically, this is the first attempt to addresses stochastic kernels using PAC-Bayes bounds. The main claims are sound. The derivation from the main claim to specific cases strengthens their usefulness.

Weaknesses: The paper is missing a thorough comparison to previous works. While the paper states previous bounds, it is unclear how the new bounds improve them, if at all. While the benefits of comparing bounds with data-dependent priors to non-data-dependent priors can be argued, it is at least meaningful to compare to previous data-dependent works. The paper might be organized differently for better readability: Previous works can be aggregated into one section, instead of being separated along with the paper; Notations and preliminaries could be defined separately outside the introduction; The implication of the two main theorems (sections 2.1-2.3) could be after the main theorems (section 3).

Correctness: The proofs inside the paper seem correct. I did not review the proofs in the appendix.

Clarity: The paper is well written. However, more organizations can be made for better readability (see details above).

Relation to Prior Work: The authors provide a comprehensive survey of previous works and explain how their assumptions differ from the previous works. Yet, a comparison between the results is still needed to put the bounds in context.

Reproducibility: Yes

Additional Feedback: In theorem 2, please better explain why the convexity is necesary and if and where it limits the general claim. Very minor issues: - Please explain the difference between “kl” (lower case) and KL (upper case) in the preliminaries. - Line 327: You wrote “or results” instead of “our results.”

[Author Response · NeurIPS 2020]

We warmly thank the reviewers for their time and for sharing valuable feedback, we will use it to improve our paper.
Overall, we are encouraged by the reviewers' reactions. Next we summarize our contributions at a high level, and then
we address each reviewer's comments separately. We hope that our responses clear out any remaining concerns.

This paper should be regarded as a theoretical contribution that takes a critical stand on the "usual assumptions" on
which PAC-Bayes inequalities are based (i.e. (a) bounded loss, (b) data-free prior, and (c) i.i.d. data observations),
clarifying their role and illustrating how to obtain PAC-Bayes inequalities in cases where these assumptions are
removed. Importantly, our work enables new PAC-Bayes inequalities with data-dependent priors. Furthermore, our
paper contributes a unified approach to understanding PAC-Bayes inequalities, and their distinctions. Our paper also
makes a case for the usefulness of formalizing data-dependent distributions as stochastic kernels, which may bring a
new perspective to the PAC-Bayes literature, in spite of the fact that stochastic kernels are well-known in other literature,
namely, on stochastic processes and their applications. We did also aim to write an informative and easy-to-read paper
that may help the larger machine learning community to connect with the PAC-Bayes literature.

## Response to Reviewer 1

We are grateful for your comments on the notation, we gladly will address this in the revised paper. Also we appreciate
your feedback about the need to elaborate more on the novelty and relevance of our paper. We will add discussions
about the new kinds of results one can obtain now that were not possible before, including rates where relevant. We will
add a remark that there is indeed a direct relationship between moments of the loss and concentration of eigenvalues of
the covariance matrix in the least-squares example. Consider a simple noise-free linear regression $Y = \mathbf{X}^\top \boldsymbol{w}^\star$, a "ridge"
prior $p^0(\boldsymbol{w}) \propto e^{-\lambda \|\boldsymbol{w}\|^2}$, and let $\hat{\boldsymbol{\Sigma}}_0 = \frac{1}{n} \sum_i \mathbf{X}_i \mathbf{X}_i^\top - \mathbb{E}[\mathbf{X}_1 \mathbf{X}_1^\top]$. Then, for the log-exponential moment of the loss
we have the identity $\log \mathbb{E} \int e^{L(\boldsymbol{w}) - \hat{L}_S(\boldsymbol{w})} p^0(\mathrm{d}\boldsymbol{w}) = \frac{d}{2} \log(2\pi) + \log \mathbb{E} |\det(\hat{\boldsymbol{\Sigma}}_0 + \lambda \mathbf{I})|^{-\frac{1}{2}}$ which shows equivalence of
concentration of eigenvalues of $\hat{\boldsymbol{\Sigma}}_0$ and concentration of the loss since $\lambda_i(\hat{\boldsymbol{\Sigma}}_0) \to 0$ as $n \to \infty$ for i.i.d. instances.

## Response to Reviewer 2

We will do a better job at discussing the usual formulation (for all posterior distributions) versus the formulation in
terms of a stochastic kernel. Their equivalence can be illustrated by discussing specific stochastic kernels. Note that
each theorem holds for an arbitrary kernel, so in that sense it holds "for all posteriors" as per the usual formulation
in the literature. On the other hand, as we mentioned in our paper, we think that it is important to make explicit the
data-dependence of the distributions, which is what the stochastic kernel formulation does. We will add a comment that
the mathematical proof of our theorem is valid for (convex) $F : \mathbb{R}^k \to \mathbb{R}$ with an arbitrary $k$, while $k = 2$ is relevant
for the known PAC-Bayes bounds. However, applications with $k > 2$ might emerge in the future, which is why we
think it is interesting to point out our theorem's validity for arbitrary $k$, as this could enable new results. We are aware
that the refined form of Pinsker inequality ($\mathrm{kl}(p\|q) \geq (p - q)^2/(2q)$, for $p < q$) is stronger than the other one only
when $q < 0.25$, we apologize for the imprecision, this will be clarified in the revised paper. Thanks for pointing out the
missing reference Blanchard & Fleuret (2007), we will make sure to include it in the revised paper.

## Response to Reviewer 3

Many thanks for your positive reaction to our work! Responding to the four raised points regarding the related literature:
(1) & (3) We will discuss in the introduction the work of Catoni (2007) in connection with data-dependent priors
(regular conditional probability distributions), and accordingly instead of claiming "firsts" we will highlight instead
that our work makes a case for the usefulness of representing data-dependent distributions as stochastic kernels, with
attribution to Catoni (2007) and Alquier (2008) as predecessors. (2) We will comment on our main theorem being valid
under general data-generation assumptions (i.e. not restricted to (c) i.i.d. data). (We are aware of this, sadly we missed
commenting on it, but happy to fix it.) We'll expand the coverage of literature on PAC-Bayes bounds for non-i.i.d. data,
including Alquier & Wittenberger (2012). The pointer to Rio's inequality is most helpful, we really appreciate this!
(4) We meant that in some cases we can have concentration of the smallest eigenvalue of a sample covariance matrix
even for unbounded instances, our reference is Section 5.4.2 of Vershynin (2011), discussing eigenvalue concentration
of a sample covariance matrix with heavy-tailed observations. We will add discussion about Holland (2019) too.

## Response to Reviewer 4

We'd like to clarify that our Theorem 2 is not restricted to convex losses, only the function $F$ is restricted to be convex
(e.g. $F(x, y) = c(x - y)^2$ was used with $x = \hat{L}_S(h)$ and $y = L(h)$), but the theorem is valid e.g. for the 01 loss or for
the ramp loss, which are non-convex. We are glad for the question, we will emphasize this in the revised paper. Also,
the notation "KL" (upper-case) is the for KL divergence in the generic case (between any pair of distributions), while
"kl" (lower-case) is reserved for Bernoulli distributions. We think this notation is helpful as a visual aid, which has been
used by other authors as well, but of course we can insert reminders throughout the paper to help the reader.

[Meta-Review · NeurIPS 2020]

This paper presents new PAC-Bayesian risk bounds that remove some of the assumptions/limitations of traditional PAC-Bayesian analysis. Namely, the new bounds allow for data-dependent priors (which have recently been shown to drastically improve tightness) and unbounded loss functions (which arise in, e.g., regression problems). While prior work has addressed both of these issues individually, the paper presents a general framework that, when instantiated, improves on some prior bounds (e.g., better constants for data-dependent priors). The paper also clarifies the distinction between "data-dependent" and "distribution-dependent" priors, which may help the broader ML community better understand these concepts. The reviews are somewhat mixed, but lean toward acceptance. The reviewers do highlight some key concerns, which I hope the authors will address when revising the paper. 1) The bounds are somewhat non-standard in that they assume a fixed learning algorithm (in this case, a kernel) that, given a dataset, maps to a posterior distribution on the hypothesis space. The standard PAC-Bayesian theorem states that, "w.h.p. over draws of data, <bound> holds simultaneously for all posteriors." The paper's main bound says that, "for any stochastic kernel, w.h.p. over draws of data, <bound> holds for the resulting posterior." As R2 points out, this does not hold uniformly for _all_ posteriors; it holds for any posterior that can be output by the kernel. I suspect that this is a weaker statement than traditional PAC-Bayes bounds. The reviewers and I would like the authors to acknowledge and discuss this in the paper. 2) As R1 points out, and the authors agree, "there is a relationship between the concentration of the loss and the concentration of the covariance eigenvalues." Since prior work has approached unbounded losses by assuming concentration of the loss, this may weaken the novelty of the results. The authors should at least discuss this in the paper. 3) Some relationships to prior work could better addressed (see reviews for specific citations). Beyond the above, I strongly the encourage the authors to incorporate _all_ feedback from the reviewers when revising the paper.